# Pathogenetic Contributions and Therapeutic Implications of Transglutaminase 2 in Neurodegenerative Diseases

**DOI:** 10.3390/ijms25042364

**Published:** 2024-02-17

**Authors:** Jun Liu, M. Maral Mouradian

**Affiliations:** RWJMS Institute for Neurological Therapeutics and Department of Neurology, Rutgers-Robert Wood Johnson Medical School, Piscataway, NJ 08854, USA; jl2574@rwjms.rutgers.edu

**Keywords:** transglutaminase 2, neurodegenerative disorders, protein crosslinking, Alzheimer’s disease, Parkinson’s disease, Huntington’s disease

## Abstract

Neurodegenerative diseases encompass a heterogeneous group of disorders that afflict millions of people worldwide. Characteristic protein aggregates are histopathological hallmark features of these disorders, including Amyloid β (Aβ)-containing plaques and tau-containing neurofibrillary tangles in Alzheimer’s disease, α-Synuclein (α-Syn)-containing Lewy bodies and Lewy neurites in Parkinson’s disease and dementia with Lewy bodies, and mutant huntingtin (mHTT) in nuclear inclusions in Huntington’s disease. These various aggregates are found in specific brain regions that are impacted by neurodegeneration and associated with clinical manifestations. Transglutaminase (TG2) (also known as tissue transglutaminase) is the most ubiquitously expressed member of the transglutaminase family with protein crosslinking activity. To date, Aβ, tau, α-Syn, and mHTT have been determined to be substrates of TG2, leading to their aggregation and implicating the involvement of TG2 in several pathophysiological events in neurodegenerative disorders. In this review, we summarize the biochemistry and physiologic functions of TG2 and describe recent advances in the pathogenetic role of TG2 in these diseases. We also review TG2 inhibitors tested in clinical trials and discuss recent TG2-targeting approaches, which offer new perspectives for the design of future highly potent and selective drugs with improved brain delivery as a disease-modifying treatment for neurodegenerative disorders.

## 1. Introduction

Transglutaminases (TGs) are a family of multifunctional enzymes that mainly catalyze the post-translational modification of target proteins or peptides through the transamidation of available glutamine residues to form inter- and intramolecular Nε (γ-glutamyl)-lysine isopeptide bonds, which are covalent, stable, and resistant to proteolysis [1]. The TG family consists of nine members, eight of which are catalytically active in humans (TG1–7 and coagulation Factor XIIIa subunit (plasma TG), and the inactive erythrocyte membrane protein band 4.2 [2]). Within this family, transglutaminase 2 (TG2) is the most extensively studied member. In addition to its Ca^2+^-dependent isopeptidase activity, TG2 exhibits additional enzymatic activities that do not require Ca^2+^, including ATP and GTP hydrolysis to mediate signal transduction through G-protein-coupled receptors. The activity of TG2 is tightly controlled by both the redox status and calcium ions. TG2 is ubiquitously expressed both inside and outside the cell and is involved in several cell biological phenomena, including cell proliferation, differentiation, and cell death [3]. In mammals, TG2 is distributed throughout the body, including the extracellular matrix and intracellular compartments of all tissues, and is implicated in several diseases, including celiac disease, cancer, fibrosis, and neurodegenerative diseases [4,5,6,7,8,9].

Neurodegenerative disorders, such as Alzheimer’s, Parkinson’s, dementia with Lewy bodies, and Huntington’s disease, are characterized by the accumulation of specific protein aggregates, inclusion body formation, and increased transglutaminase activity in affected brains [10,11,12,13]. The aggregated proteins in these afflicted brains, amyloid-beta, tau, α-synuclein, or huntingtin, are all substrates of TG2 through its catalytic crosslinking activity [5]. The increased understanding of the involvement of TG2 in these various neurodegenerative diseases points towards an attractive therapeutic potential, prompting the need for the design and identification of TG2 inhibitors. Different strategies can be employed to target TG2, including inhibiting its enzymatic transglutaminase activity, decreasing its expression, or modulating its trafficking. Already, a TG2 inhibitor has been studied with promising results in celiac disease [14,15]. However, designing highly selective inhibitors with a favorable pharmacokinetic profile and access through the blood–brain barrier remains a challenge. In this review, we discuss the molecular regulation and pathogenic role of TG2 in neurodegenerative diseases, as well as the latest efforts in the development and optimization of TG2 inhibitors.

## 2. Structure and Multifunctional Activities of TG2

The human *TG2* gene localizes to chromosome 20q11-12, and its exons span approximately 37 kb. In addition to the more abundant full-length TG2, which contains 687 amino acids and has a molecular weight of approximately 78 kDa, four different isoforms generated by alternative splicing have been identified, including TGM2_V1, V2, V3, V4a, and V4b [16]. The main feature of these isoforms is the truncation of the C-terminus of different lengths, including the lack of the GTP-binding regulatory domain, which controls the response of TG2 to Ca^2+^ activation [17]. The short splice variants of TG2 have been detected in a variety of cells, such as astrocytes, neurons, and endothelial cells [18]. Interestingly, the preferential expression of the splice variants TGM2_V4a and v4b compared to full-length TG2 was recently reported in peripheral blood mononuclear cells derived from patients with primary progressive multiple sclerosis (PP-MS), suggesting that TG2 splice variants may function in the pathophysiology of PP-MS [19]. However, the differential regulation of these variants remains unclear.

The secondary structure of full-length TG2 consists of four domains (Figure 1a) [20]: the N-terminal β-sandwich domain (aa 1–139), which has a fibrontectin-binding site [21,22]; the core domain (aa 140–460), which contains a catalytic triad and Ca^2+^-binding sites [23]; and the two C-terminal β-barrel domains (aa 461–586 and aa 587–687), which include GTP/GDP binding [24] and phospholipase C-binding sites [25]. Within the core domain, the catalytic triad Cys (277)-His (335)-Asp (358) is necessary for TG2 transamidation activity, and two tryptophan residues, Trp (241) and Trp (332), participate in the stabilization of the thioester intermediate (Figure 1a). Of these residues, Cys277 is the essential nucleophile for transamination, and the mutation of this cysteine to serine (TG2-C277S) has been extensively used experimentally as a catalytically inactive TG2 mutant [26,27]. However, C277S mutation is also deficient in the GDP/GTP binding ability of TG2 [28], thus limiting the utility of this mutant for examining the specific role of TG2 transamidation activity in numerous biological processes. To this end, mutating W241 in the enzymatic core to an alanine (W241A) specifically abolishes the crosslinking enzymatic activity of TG2 while preserving its GTPase function [29] and provides a preferred genetic tool to study specifically its transamidation activity in vitro and in vivo [27,30].

TG2 is well characterized regarding its Ca^2+^-dependent crosslinking and transamidation activities, which allow it to modify substrate proteins post translationally via the formation of covalent bonds [31]. Specifically, a cysteine residue located in the active site of TG2 attacks γ-carboxamide groups in a specific protein glutamine residue (acyl donor) to yield a γ-glutamyl thioester. The latter subsequently reacts with nucleophilic primary amines, leading to the formation of either a (γ-glutamyl) amine bond or a covalent isopeptide bond, which is resistant to degradation (Figure 1c) [3,32]. These reactions affect the conformation of proteins and trigger the formation of stable, rigid, and insoluble supramolecular complexes [33]. TG2 also has the ability to catalyze the hydrolysis of guanosine triphosphate (GTP) as a G-protein signal transduction protein that mediates intracellular signaling (Figure 1d). Under physiological conditions, TG2 exerts additional enzymatic activities that do not require Ca^2+^; these include protein disulfide isomerase and serine/threonine protein kinase [34]. Moreover, interacting with several other proteins as an adhesion or scaffold protein is a crucial non-enzymatic activity of TG2, especially in the extracellular environment. The importance of TG2 as a scaffold protein has been reported recently in various physiological or pathological settings [35,36,37,38]. For example, TG2 has been shown to negatively regulate the cyclic GMP-AMP synthase (cGAS)–stimulator of interferon genes (STING) signaling pathway by impairing interferon regulatory factor 3 (IRF3) phosphorylation independent of its transamidase activity in human melanoma cells, suggesting a role of TG2 in reducing type I interferon (IFNI) production after DNA damage to limit the immune system response in cancer cells [37]. The cGAS-STING axis is one of the key signaling pathways activated in the presence of both self and pathogen DNA in the central nervous system and is a pivotal driver of chronic inflammation and functional decline during aging [39,40]. The excessive activation of this pathway induces neuronal cell death that leads to neurodegeneration and cognitive decline in aged mice [40]. Thus, targeting the cGAS-STING-pathway-mediated inflammatory response may provide a potential therapeutic strategy to suppress neurodegenerative processes [40]. However, whether TG2 directly contributes to the cGAS-STING axis in human neurodegenerative diseases or age-related inflammation remains to be explored.

## 3. Regulation of TG2 Expression and Its Activation

### 3.1. Regulation of TG2 Transcription

As a multi-functional protein, TG2 is highly regulated at many levels [41,42]. The promoter of the *TG2* gene contains a number of critical response elements, including for retinoid acid, NF-κB, IL-6, TGFβ1, and hypoxia-inducible factor 1 (HIF1) [42]. Among these molecules, retinoid acid [43] is a highly important regulator of TG2 expression in a broad range of cell types and species [41,44]. NF-κB and HIF1 are two of the oxidative-stress-responsive transcription factors that regulate TG2 expression [45]. There are six putative hypoxia response elements in the promoter of the *TG2* gene [18]. HIF1, which is a heterodimeric complex consisting of inducible HIF1α and constitutively expressed HIF1β, targets the *TG2* promoter in response to hypoxia [46]. Notably, increased expression of TG2 and its activity is a common feature of increased ROS and several inflammatory diseases [45,47,48]. NF-κB plays a pivotal role in inflammation and is activated under diverse oxidative stress conditions. TG2 transcription increases in response to cytokines such as TNFα and IL-6 via the NF-κB-dependent signaling pathway induced by various oxidative stimuli [45]. Interestingly, TG2 is known to activate NF-κB through crosslinking, and it polymerizes IκBα or PPARγ, implying a positive feedback loop generated by NF-κB and TG2 causing constitutive NF-κB activation [49,50]. Moreover, TG2 mRNA and protein levels are increased in rat astrocytes treated with lipopolysaccharide (LPS), a reagent that induces the release of pro-inflammatory signals [51]. Notably, this effect is inhibited by an antioxidant, ethyl pyruvate, suggesting the role of oxidative stress in TG2 regulation [51]. In addition, the *TG2* promoter also contains a TGF-β response element. The interplay between TGF-β1 and TG2 is intricate, as TGF-β1 can have both activating and inhibitory effects on TG2 depending on the context and the specific cellular and molecular environment [3,52,53]. And TGF-β expression is reported to down-regulate inflammatory and autoimmune responses [54,55], suggesting the contribution of TG2 in the neuroinflammation that accompanies neurodegenerative disorders [56].

### 3.2. Regulation of TG2 Activation

Under physiological conditions, TG2 catalytic activities are tightly controlled by multiple factors including Ca^2+^, guanine nucleotides, and the redox environment, and are associated with specific conformational states [57,58]. TG2 exists in two conformations, referred to as open/extended (active) and closed/folded (inactive) forms, each of which has a distinct and mutually exclusive activity (Figure 1b) [58,59]. The closed conformation has GTP/GDP-binding/GTPase activity [60], while the open conformation functions as a transamidase that catalyzes the calcium-dependent formation of covalent protein crosslinking [32,34]. Under physiological conditions, Ca^2+^ levels are low (~100 nM free cytoplasmic [Ca^2+^]) and GTP levels are high (100–150 μM), the closed conformation predominates in the intracellular environment, and transamidase activity is inhibited [61,62]. When the calcium levels in the cytosol are elevated (0.5–1.5 mM) in response to extracellular stimuli, the interaction with Ca^2+^ induces a conformational change that shifts TG2 to an open form, exposing the catalytic core to allosterically activate its transamidase activity and inhibiting GTP/GDP binding (Figure 1b) [34,57]. During this conformational change, the interaction with multiple substrates or binding partners is greatly affected and results in several biological changes [57,63]. In the extracellular space, Ca^2+^ levels are present in the mM range and the GTP concentration is low, favoring TG2 activity. However, the data in cellular and in vivo models indicate that extracellular TG2 is typically inactive under normal physiological conditions and that physical or chemical injury can reversibly induce TG2 activation [64]. Mechanistically, the formation of a vicinal disulfide bond between Cys370 and Cys371 in the TG2 catalytic core keeps extracellular TG2 in the inactive state even at a high Ca^2+^ concentration [65], and this can be reversed by thioredoxin-mediated reduction [66].

Additionally, the accumulation of intracellular reactive oxygen species (ROS) caused by various cellular stresses has been shown to activate TG2 via several mechanisms. Firstly, ROS-sensitive transcription factors, NF-kB and H1F [45], are involved in the regulation of TG2 promoter activity, as described above. Secondly, TG2 is regulated by post-translational modifications, such as SUMOylation and ubiquitination, via the ROS-mediated pathway [67,68]. Thirdly, calcium chelating agents block ROS-induced TG2 activation, indicating that ROS-induced TG2 activation occurs through elevated Ca^2+^ [46,68,69]. Notably, increasing evidence suggests an interplay between calcium and ROS in various cellular processes [70]. High levels of calcium stimulate respiratory chain activity leading to excessive ROS, and ROS, in turn, target calcium channels that enhance the release of calcium [71]. Indeed, ROS–calcium crosstalk is implicated in several pathophysiological conditions, including neurodegenerative diseases [72].

Several strategies can be utilized to prevent or modulate the activation of TG2, including (1) pharmacological inhibitors that are directed towards blocking TG2 activity; (2) interventions that reduce oxidative stress or other cellular stressors and may indirectly curtail TG2 activity, such as antioxidant agents ethyl pyruvate, N-acetylcysteine (NAC), and α-Lipoic Acid [51,73,74,75]; and (3) the possible exploration of the genetic modulation of TG2 expression, including gene-silencing techniques and CRISPR/Cas9 gene editing to regulate TG2 levels [76,77,78]. However, the feasibility and safety of the latter approaches need thorough investigation [78].

## 4. Distribution of TG2

While TG2 has been identified predominantly in the cytosol, it is also detected in various subcellular compartments, including the mitochondria, nucleus, plasma membrane, and extracellular matrix.

### 4.1. Cytosolic TG2

Cytosolic TG2 mediates post-translational modifications to intracellular proteins through its crosslinking activity, which affects the biological activities of target proteins and relevant signaling events in the cell, including NF-κB signaling, inflammatory response, autophagy, vesicular trafficking, EGF-/EGFR-induced signaling, and oncogenesis [58]. For examples, TG2 modulates the serotonylation of several cytoplasmic proteins, including small regulatory GTPases RhoA and Rac1, as well as cytoskeletal components, such as α-actin, that alter the cytoskeleton and vesicular trafficking [79]. In addition, TG2 mediates IκBα polymerization, which promotes IκBα proteasomal degradation and NF-κB activation [49], leading to the regulation of the expression of multiple target genes that play roles in cell survival. Moreover, GTP-binding-defective TG2 mutants induce apoptosis in fibroblasts independently of their transamidase activity [80], suggesting that cytosolic TG2 impacts its intracellular target proteins in an activity-independent manner.

### 4.2. Mitochondrial TG2

TG2 is found in mitochondria in various cell types, especially in response to apoptotic signaling [81]. In neuroblastoma cells, mitochondrial TG2 comprises up to 50% of total cellular TG2 and binds to both the outer and inner mitochondrial membranes [82]. Interactome analysis has identified the interaction between TG2 and glucose-regulated protein 75 (GRP75), which occurs in mitochondria-associated membranes (MAMs). GRP75 acts as a bridging protein interconnecting two organelles by assembling the IP3R1–GRP75–VDAC1 complex, which enhances the ER–mitochondria interaction and modulates ER–mitochondria Ca^2+^ transfer. The absence of TG2-GRP75 binding decreases the number of ER–mitochondria contact sites and impairs Ca^2+^ flux between these two organelles, highlighting the critical regulatory role of TG2 in MAMs [83]. Moreover, TG2 interacts with the proapoptotic protein Bax and is involved in BAX translocation to the mitochondria, inducing the apoptotic pathway and thus supporting the role of TG2 in apoptosis [84]. Considering that TG2 crosslinking activity is calcium-dependent, this function does not typically occur in normal tissues, whereas it is likely to be activated in the context of abnormally elevated calcium levels in mitochondria in disease states, such as cerebrovascular ischemia/reperfusion injury, Huntington’s disease, and other neurodegenerative disorders [42].

### 4.3. Nuclear TG2

Nuclear TG2 accounts for about 5~7% of total intracellular TG2 [18]. There are three putative nuclear localization signals and one putative leucine-rich nuclear export sequence in the primary sequence of TG2 (Figure 1a), and these provide the possibility for TG2 to shuttle between the cytoplasm and nucleus [85,86]. The translocation of TG2 to the nucleus and its activation usually occur under a variety of stress stimuli, such as increased Ca^2+^ levels, hypoxia, and exposure to retinoic acid [87,88,89]. Nuclear TG2 is known to mediate crosslinking of the transcription factor Sp1, which results in the activation of caspase-mediated apoptosis [90]. The TG2-catalyzed serotonylation and dopaminylation of histone H3 has been reported, suggesting the contributions of TG2 in regulating the genome and transcription [91]. Additionally, several histones, including four core histones, H2A, H2B, H3, and H4, have been described as substrates of TG2 transamidation, and their crosslinking mediates the apoptosis-induced condensation of chromatin [91,92,93,94]. A recent study illustrates that the TG2 transamidation activity on histone is regulated by its substrate steric accessibility [95]. In a Huntington’s disease (HD) cellular model, the hyperactivation of TG2 in the nucleus has been shown to result in abnormal histone H3 polyamination to modulate facultative heterochromatin formation [96,97]. As a consequence, genes such as peroxisome proliferator-activated receptor-γ (PPARγ), coactivator 1α (PGC1α), and cytochrome *c* (Cyt *c*), which are critical for mitochondrial biogenesis, are dysregulated, suggesting that nuclear TG2 contributes to the dysfunctional mitochondria in this disease [96,97].

### 4.4. Extracellular TG2

TG2 can be released into the extracellular environment, where it is associated with, and covalently modifies, the post-translational modification of extracellular matrix proteins, including fibronectin, laminin, vitronectin, fibrinogen, and collagen [34]. For example, fibronectin and collagen fibrils have been shown to be covalently modified by TG2, leading to the formation of large, multimeric complexes with high chemical stability [98,99]. The increased rigidity of fibronectin results in enhanced osteoblast and fibroblast adhesion [100,101]. This TG2-mediated crosslinking enhances the stabilization and stiffening of the extracellular matrix structure [102], which can affect cell behavior in health and disease [103,104]. In addition, integrins are important transmembrane adhesion and signaling receptors that regulate intracellular signaling pathways. TG2 binds strongly to both integrins and fibronectin, enhancing their weak interaction, facilitating cell adhesion to the matrix, and activating integrin-mediated signaling. Indeed, this extracellular integrin/TG2/fibronectin binding is important in various disease conditions, such as cancer cell metastasis and glial scarring [105,106]. Moreover, increased extracellular TG2 plays a role in the aberrant transamidation of extracellular matrix components and fibrosis progression in the kidney [7], liver [107], and heart [108,109] in disease states.

### 4.5. TG2 Expression in Various Cell Types

As mentioned above, TG2 is a multifunctional protein that is widely expressed; hence, it is crucial to examine the variations in its activity, subcellular localization, and function across different cell types. For example, TG2 is expressed in all glial cell types in the central nervous system (CNS), including astrocytes, microglia, and oligodendrocytes, but the function of TG2 varies depending on different glial cell types [110]. Mice with TG2 deletion specifically in astrocytes exhibit significant improvements in motor function following spinal cord contusion injury. Similarly, the inhibition of TG2 in wild-type mice with a TG2-specific irreversible inhibitor showed improved functional recovery after spinal cord injury [111], indicating the negative role of TG2 in modulating the response of astrocytes to CNS injury. On the other hand, microglia-derived TG2 reportedly interacts with GPR56, a member of the adhesion G-protein-coupled receptor family, in the presence of laminin, and promotes myelin formation and repair during neuronal regeneration [112].

## 5. Pathological Role of TG2 in Neurodegenerative Diseases

TG2 is expressed in the mammalian CNS, including in the human brain [10]. As the most abundant transglutaminase in the nervous system, TG2 has a differential expression in different brain regions, suggesting a complex role played by TG2 in the brain [10]. Increasing evidence indicates that the diverse functions of TG2 enable it to be involved in various biological processes in a context-specific manner. In keeping with this, up-regulated TG2 mRNA and a corresponding increase in protein level and enzymatic activity have been observed in many neurodegenerative diseases, including Alzheimer’s (AD), Parkinson’s (PD), Huntington’s (HD), and other polyglutamine expansion disorders [5,113]. In these diseases, calcium dys-homeostasis, energy depletion, and increased oxidative stress are well-known phenomena in damaged cells, contributing to the activation of TG2 (Figure 2), as reviewed above. A growing body of experimental data has been published suggesting the involvement of TG2 in the pathophysiology of all these diseases, including the deposition of specific insoluble protein aggregates, supporting the notion that aberrant TG2 activity contributes to the pathogenesis of these disorders (Figure 2).

### 5.1. TG2 in Alzheimer’s Disease

Alzheimer’s disease (AD) is the most common neurodegenerative disorder and type of dementia, and it is characterized by the deposition of neurotoxic protein aggregates, namely, extracellular amyloid-β (Aβ) in the form of plaques and intracellular hyperphosphorylated tau in the form of neurofibrillary tangles (NFTs), associated with neuroinflammation and neurodegeneration (Figure 2e,f) [114,115]. In various AD models, the release of calcium from the endoplasmic reticulum (ER) through the dysregulated ryanodine receptor (RyR) and inositol-1,4,5-trisphosphate receptor (IP3R) is exaggerated [116,117]. As a result of excess calcium release from ER stores, mitochondrial calcium overload occurs, leading to metabolic dysfunction and an increased production of ROS, which induce intracellular TG2 activity [118,119]. Aggregated Aβ can feedforward, inducing different events involved in AD pathogenesis, including the dysregulation of Ca^2+^ and mitochondrial homeostasis [120,121,122]. In in vitro models, TG2 has been shown to generate a broad range of Aβ oligomers, leading to protofibril-like assemblies [123]. The association between TG2 and Aβ forming toxic aggregates induces an acute mitochondrial Ca^2+^ overload, increased mitochondria–endoplasmic-reticulum contacts, and synaptic dysregulation in AD [124]. In APP_SWE_/PS1_ΔE9_ and APP23 mouse models of AD, TG2 protein, as well as its crosslinking activity, are colocalized with Aβ plaques and vascular Aβ in the brain [125]. In addition, the cross-breeding of the AD-mimicking APP23 mouse model with TG2 knock-out mice has been shown to result in a significant reduction in the number of senile plaques, small dense plaques, and vascular Aβ deposits in the cortex, suggesting the critical role of TG2 in Aβ deposition in AD [126,127]. Notably, in AD patients, TG2 expression and activity are significantly increased in the prefrontal cortex and cerebrospinal fluid compared to control subjects, and this finding correlates with cognitive impairment [128,129]. Additionally, increased TG2 immunoreactivity and its catalyzed crosslinks colocalize with Aβ in senile plaques, as well as in the vessel wall in early stage cerebral amyloid angiopathy (CAA), suggesting the involvement of TG2 early in the formation of these pathologic features in AD-affected brains [130,131,132].

In vitro biochemical and kinetic studies into tau have demonstrated that it is also a substrate of TG2, and that TG2 catalyzes the crosslinking of tau to produce insoluble polymers [133]. And the TG2-mediated polyamination of tau is resistant to proteolytic degradation by µ-calpain [134], suggesting the contribution of TG2 in the formation of tau aggregates in AD. In the brains of transgenic mice expressing the P301L mutant tau protein, which exhibit tau pathology, neuronal loss, glial activation, and behavioral deficits [135], TG2 activity is significantly increased compared with four-repeat wild-type tau transgenic and non-transgenic mice that do not exhibit tau pathology. Additionally, phosphorylated tau and TG2-catalyzed crosslinks are highly colocalized in the hindbrain, spinal cord, and cortex of this mouse model, suggesting that the TG2-mediated crosslinking of phosphorylated tau is involved in the formation or stabilization of neurofibrillary tangles [136]. A recent study using another mouse model of AD that overexpresses human wild-type tau detected increased TG2 mRNA and protein levels in the dorsal raphe nucleus compared to wild-type mice at a relatively young age when depressive-like behaviors were noted, suggesting a role of TG2 in tau pathology in the prodromal phase of AD prior to the onset of cognitive decline [137]. Notably, in postmortem studies on AD-affected brains, TG2- and TG2-catalyzed covalent isodipeptide bonds co-localize with tau-positive NFT in the hippocampus and neocortex [130,138]. Interestingly, isodipeptide bonds reportedly co-localize with paired helical filaments (PHFs) in the parietal and frontal cortex, lacking microscopically detectable NFT in stage II AD, suggesting that the TG2 catalyzed crosslinking of tau is an early event in the formation of PHF, prior to the appearance of NFT in AD [139].

### 5.2. TG2 in Parkinson’s Disease and Dementia with Lewy Bodies

Parkinson’s disease (PD) is the second-most-common neurodegenerative disorder after Alzheimer’s. The report from the World Health Organization in 2022 highlighted that the global impact of PD is increasing faster than that of any other neurological disorder [140]. PD is characterized by the accumulation of α-Synuclein (α-Syn) into insoluble cytoplasmic inclusions called Lewy bodies and Lewy neurites, associated with progressive neuronal death, most notably in the substantia nigra pars compacta through complex molecular and cell biologic insults, including mitochondrial impairment, lysosomal dysfunction, and calcium dys-homeostasis [141]. In PD-affected brains, the expression of Ca_v_1.3-type voltage-gated calcium channels was seen to be increased in substantia nigra neurons compared with aged non-PD controls, leading to increased cytosolic calcium and mitochondrial oxidative stress, which promoted TG2 activation [142].

Accumulating evidence has shown that α-Syn is a target of TG2 transamidation activity in vitro and in vivo [30,143,144,145,146], implicating TG2 activity in α-Syn aggregation and the pathogenesis of synucleinopathies (Figure 2g). In a neuroblastoma SH-SY5Y cellular model, increased TG2 activity and TG2-dependent α-Syn crosslinking were detected following exposure to the dopaminergic toxin 1-methyl-4-phenylpyridine (MPP (+)), while the pharmacological blockade of TG2 with Z006 was found to inhibit this effect [147]. In human postmortem studies, TG2 mRNA expression, protein level, and activity are increased in the substantial nigra of PD-affected brains, and TG2-mediated ε-(γ-glutamyl) lysine bonds (GGEL) bonds co-localize with α-Syn in Lewy bodies, and the two proteins co-immunoprecipitate in extracts of PD substantia nigra [144,148]. To elucidate the sequence and structural basis of the crosslinking activity of TG2 targeting α-Syn, Gln109 and Gln79 were identified as the primary TG2 reactive sites [149]. Additionally, phosphatase and tensin homologue-induced putative kinase 1 (PINK1), which is linked to recessively inherited early onset PD, reportedly phosphorylates TG2 and inhibits its ubiquitination and proteasomal degradation, leading to an increased TG2 accumulation and crosslinking effect [150]. Accordingly, TG2 activity is decreased in PINK1 knock-out mouse brains [150]. These observations suggest a putative regulatory role played by PINK1 in TG2-mediated cellular protein dys-homeostasis and inclusion formation [150,151].

Notably, in vivo studies in genetically modified mice that are transgenic for α-Syn and either over-express TG2 (TG2^Tg^/Syn^Tg^) or have deletion of the TG2 gene (TG2^−/−^/Syn^Tg^) support the contribution of TG2 to the pathogenesis of PD by crosslinking α-Syn [30,145,146]. Specifically, high-molecular-weight species of α-Syn and proteinase K-resistant α-Syn aggregates are increased in TG2^Tg^/Syn^Tg^ mouse brains when compared with Syn^Tg^ mice [145], while TG2^−/−^/Syn^Tg^ mice exhibit the reverse [146]. An exaggerated neuroinflammatory response and increased neuronal damage are seen in TG2^Tg^/Syn^Tg^ mice, whereas these pathologic features are attenuated in TG2^−/−^/Syn^Tg^ mice, further supporting that TG2-mediated α-Syn crosslinking exacerbates α-Syn toxicity. Accordingly, targeting TG2 activity might be a plausible strategy to prevent the pathologic aggregation of α-Syn and its toxicity, and, consequently, the development of PD.

### 5.3. TG2 in Huntington’s Disease

Huntington’s disease (HD) is a hereditary, fatal neurodegenerative disorder characterized by extensive neuronal loss, particularly in the striatum and cerebral cortex, clinically characterized by involuntary movements, cognitive impairment, and psychiatric manifestations [152,153]. HD is caused by the expansion of a CAG trinucleotide repeat in the gene encoding for the protein huntingtin (HTT). In humans, the *HTT* gene normally contains between 6 and 35 CAG repeats, whereas HD is caused by a mutation in the *IT-15* gene that expands a highly polymorphic CAG tract with greater than 39 repeats [153]. As a consequence, the mutated protein HTT (mHTT) contains disease-causing expansions of glutamines that make it prone to misfold and aggregate [154]. These aggregates are found in neuronal intranuclear inclusions and dystrophic neurites [155] and represent a pathological hallmark of HD [156]. The *mHTT* gene interferes with a variety of transcriptional targets involved in calcium homeostasis and signaling, including the up-regulation of calretinin, presenilin 2, calmyrin 1, huntingtin-associated protein 1, and calcyclin-binding protein [157]. The mHTT protein interacts with IP3Rs, resulting in exaggerated calcium release from ER stores [158]. The increase in cytosolic calcium concentration drives ROS production, leading to TG2 activation [159]. Accordingly, the activation of TG2 may be an early and significant consequence of HD, since mHTT promotes Ca^2+^ elevations.

In the frontal cortex of postmortem HD brains, TG2 colocalizes with both mHTT protein and TG2-catalyzed covalent GGEL bonds in intranuclear inclusions (Figure 2h) [160]. Calmodulin reportedly enhances TG2 activity, leading to the crosslinking of mHTT fragments and the formation of insoluble high-molecular-weight aggregates [161]. Additionally, TG2 enzymatic activity is increased in neuronal nuclei in HD patient brains [162], and the products of TG2 crosslinking activity are increased in HD cerebral spinal fluid compared with that of control subjects [163].

The critical role of TG2 in aggravating HD pathogenesis is demonstrated through gene transcriptional regulation. Normal HTT protein binds to actin and is required for intranuclear cofilin–actin rod formation and clearance during the stress response. This process is disrupted by the polyglutamine expansion of HTT. In the presence of the mHTT protein, nuclear actin remodeling is defective, characterized by fewer but longer-persisting cofilin–actin rods, leading to faster cell death correlating with HD progression [164,165]. Notably, aberrant TG2 activity in HD induces the increased crosslinking of actin and cofilin in the presence of mHTT, leading to the persistence of cofilin–actin rods, which results in cell death [164]. Interestingly, nuclear cofilin–actin is required for RNA polymerase II transcription elongation and, therefore, impacts transcriptional regulation. Thus, the accumulation of nuclear cofilin–actin rods may impair gene expression [166,167]. Moreover, studies in cellular and drosophila models have shown that TG2 acts as a selective co-repressor of the nuclear genes involved in HD, whereas the inhibition of its activity or the prevention of its translocation to the nucleus fails to repress transcription [97]. In addition to TG2, TG6 physically interacts with mHTT and contributes to mHTT aggregates in HD transgenic animals, highlighting the role of TG6 activity in the pathogenesis of HD [168].

### 5.4. TG2 in Other Neurodegenerative Disorders

In addition to AD, PD, and HD, TG2 is implicated in the pathogenesis of other neurodegenerative diseases, including amyotrophic lateral sclerosis (ALS) and multiple sclerosis (MS).

ALS is characterized by the selective degeneration and dysfunction of motor neurons. Mutations in the superoxide dismutase 1 (*SOD1*) gene account for 10–20% of familial cases and 1–2% of sporadic cases [169,170,171]. A significant elevation in serum transglutaminase activity correlating with disease severity in the early stages of ALS, followed by depletion in later stages, has been reported and attributed to extensive spinal motor neuron death with disease progression. The depletion of transglutaminase activity in the cerebrospinal fluid in later stages is also reported. These findings suggest a potential role of TG in ALS pathogenesis [172]. Additionally, in a mouse model of ALS, TG2 interacts with misfolded SOD1 (mSOD1) protein and is involved in mSOD1 oligomerization, contributing to neuroinflammation and disease progression [173].

Multiple sclerosis (MS) is a relapsing–remitting or primary progressive disease characterized by neuroinflammation, demyelination, and, eventually, axon loss and neurodegeneration in the CNS. Accumulating evidence from studies has identified the involvement of TG2 in postmortem MS brain samples and in the central nervous system of the animal model for experimental autoimmune encephalomyelitis (EAE), illustrating its role in the pathogenesis of MS [174,175,176,177,178]. For example, TG2 knock-out mice subjected to the EAE model exhibited decreased disease severity compared to wild-type littermates, and treatment with the TG2 inhibitor cysteamine in the EAE effector phase in wild-type mice showed benefits [179]. In a rat EAE model of MS, the inhibition of TG2 activity with an irreversible TG2 inhibitor, KCC009, reduced monocyte infiltration into the CNS accompanied with an ameliorated phenotype [180]. TG2 is increased in MS patient-derived monocytes, affecting their adhesion and migration into the CNS [176]. In keeping with this, elevated TG2 mRNA levels have been reported in peripheral blood mononuclear cells in MS patients, a finding that correlated with disease progression [176,177], suggesting TG2 as both a biomarker and therapeutic target for MS.

## 6. Current Progress in Targeting TG2 for Therapeutic Purposes

As described above, accumulating evidence shows that TG2 has multifunctional roles and that the dysregulation of TG2 is implicated in the pathogenesis of various neurodegenerative diseases, making it an attractive therapeutic target in these disorders [181]. The pharmacological inhibition of TG2 represents a plausible strategy to modulate intracellular TG2 activity. Given that the pathogenic role of TG2 is linked primarily to its crosslinking activity, targeting TG2 for therapeutic purposes has focused on specifically blocking its transamidation activity. To date, three main types of TG2 inhibitors have been developed based on their mechanisms of inhibition (Figure 3): competitive amine inhibitors, reversible inhibitors, and irreversible inhibitors [182,183]. Specifically, competitive amine inhibitors occupy the active site of TG2 by competing with its amine substrates, such as protein-bound lysine residues, to inhibit the transamidation activity of TG2; reversible inhibitors block substrate access to the active site of TG2 without covalent modification, whereas irreversible inhibitors inactivate TG2 by covalently modifying it, thereby preventing its substrate binding [184].

### 6.1. Competitive Amine Inhibitors

The earliest TG2 inhibitors were of this type (Figure 3a). Monodansyl cadaverine (MDC) is one of the widely used competitive inhibitors of TG2 [185]. It has shown protective activity against insoluble aggregate formation for CAG repeat expansion disease proteins [186]. However, the non-specificity of this amine limits its therapeutic utility.

Cystamine and its reduced form, cysteamine, are commonly used as inhibitors of TG2 by different mechanisms [187] and have been tested as potential therapeutics for a broad range of diseases, including neurodegenerative disorders [188,189]. Cystamine has been demonstrated to irreversibly inhibit human TG2 transamidation activity by promoting the formation of the disulfide bond between Cys370 and Cys371 [188], whereas cysteamine acts as a competitive amine inhibitor to target cysteine residues in the active site for transamidation reactions catalyzed by TG2 [187]. Cystamine has shown protective effects in in vivo models of HD and PD, while cysteamine was recently shown to protect neurons against the toxicity mediated by mHTT in primary neurons and iPSC models of HD [190], supporting the conduct of clinical trials in these diseases. However, the specificity of cystamine for TG2 has been questioned since it has the ability to broadly target transglutaminases and inhibit caspases [191], indicating the possibility of off-target effects from this agent. Moreover, although cysteamine has been shown to be well tolerated in patients with HD [192,193], adverse effects, such as rashes, fever, nausea, and motor impairment were observed in HD patients using cysteamine in the CYTE-I-HD clinical trial [192], and asthenia or fatigue was more frequently observed in HD phase II trials [193], limiting its clinical utility.

### 6.2. Reversible Inhibitors

Several reversible inhibitors of TG2 have been identified (Figure 3b). For example, compound CP4d was identified as a potent cinnamoyl triazole inhibitor with an IC50 value of 2.1 µM [194], which competes with the acyl-donor substrate binding site for transamidation activity and suppresses TG2’s binding ability to GTP [195]. CP4d is widely used in various cellular models to explore the functions of TG2. For example, CP4d reduces cancer stem cell survival, migration, and invasion [196], suggesting that TG2 is a cancer cell survival protein [197].

One particularly interesting reversible inhibitor compound, LDN-27219, was designed to bind to the GTP-binding pocket of TG2, which promotes TG2 in its closed conformation and inhibits transamidase activity in mesenteric arteries [198]. Stabilizing the closed conformation of TG2 with LDN-27219 has been demonstrated to be more effective than irreversible TG2 inhibitors such as Z-DON in reducing blood pressure in Wistar Hannover rats [199]. These findings suggest that promoting the closed TG2 conformation may serve as a potential strategy to treat age-related vascular dysfunction and lower blood pressure. Recently, TG2 inhibition with LDN27219 in colorectal cancer cells was reported to prevent cell proliferation and tumorsphere formation, and blocking TG2 with LDN27219 in vivo exhibited a significant inhibition of tumor progression [200]. Unlike LDN-27219, TTGM-5826 was designed in an opposite manner to stabilize TG2 in its open form. The compound was shown to inhibit the growth and migration of various cancer cells with high TG2 expression and exhibited selectivity between transformed and healthy cells [201]. With further development, TTGM-5826 analogues might become potential anti-cancer treatment candidates.

### 6.3. Irreversible Inhibitors

Currently, most medicinal chemistry efforts targeting TG2 focus on developing highly potent irreversible inhibitors (Figure 3c), in which certain electrophilic functional groups that react efficiently with the active site of TG2 have emerged as privileged ‘warheads’, including acrylamides, dihydroisoxazoles, thiadiazoles, and epoxides. In addition to its chemical properties, the warhead must be designed to keep its stability in cellular conditions and prevent off-target effects [202,203]. A commonly used irreversible inhibitor, Z-DON (Z006), has been used to demonstrate the modulation of TG2 in neurite outgrowth and neural differentiation when neurons are exposed to organophosphates [204]. Most recently, Z-DON, reduced molecular events, such as mitochondrial membrane potential loss and synaptic failure, caused by Aβ toxic aggregates in a transgenic model of AD [124]. The Khosla group has synthesized a large series of dihydroisoxazole-based irreversible inhibitors. Among these compounds, KCC009 blocks TG2-mediated fibronectin remodeling in both in vitro and in vivo studies [205,206] and increases sensitivity in mice harboring glioblastomas to carmustine chemotherapy, leading to improved survival [205]. However, the low aqueous solubility of KCC009 limits its clinical utility [184]. Another compound, ERW1041E, was found to inhibit TG2 activity in the small intestine in mouse models of celiac disease and to be well tolerated by mice at concentrations up to 50 mg/kg [207]. But it shows poor affinity, with a minimum TG2 inhibitory concentration of 6–12 µM in vitro [208]. NC9 is yet another effective, irreversible TG2 inhibitor that shifts the enzyme from its closed conformation to open conformation and prevents both GTP signaling and transamidation activity [196]. The inhibition of TG2 activity with NC9 reduces the ability of SCC-13 cell-derived epidermal cancer cells to form tumors and inhibits their migration and invasion [209]. NC9 has also been found to be efficacious in reducing proliferation in a large subset of glioblastoma cell types and exhibited no toxicity to neurons, suggesting that TG2 plays a context-specific role in glioblastoma [210].

To date, the most important irreversible inhibitor produced by Zedira, ZED1227, has completed a phase II clinical trial for the treatment of celiac disease, with improved symptom and quality-of-life scores [14,15]. ZED1227 shows high potency, with an IC50 of 45 nM and specificity for targeting TG2, but its limitation to cross the blood–brain barrier (BBB) precludes its utility for neurodegenerative diseases [30,211]. More recent efforts in the design or optimization of novel irreversible inhibitors serve as a template for the future development of potent therapeutic tools [202,203,212,213].

## 7. Conclusions

Overall, a substantial body of evidence has accumulated pointing to the pathogenetic roles of TG2 in several aspects of neurodegenerative diseases, including the formation of protein aggregates, neuroinflammation, and the regulation of cell death. Thus, inhibiting TG2 may prevent all these pathogenetic features of neurodegenerative disorders. These potential benefits make TG2 a compelling therapeutic target and have encouraged the identification of many TG2 inhibitor compounds and proof-of-concept studies in various disease models. Further research is needed to overcome the challenges to therapeutic developments for these diseases. In addition to pharmacokinetic/pharmacodynamic properties, the major challenges for TG2-based drug discovery are the specificity and brain–drug delivery. The limitations caused by the blood–brain barrier and the shortcomings of current inhibitors necessitate the identification of CNS-appropriate inhibitors. Among these approaches, nanotechnology that employs materials in nanoparticles has proven to be a safe and highly suitable drug platform for brain diseases [214,215,216]. Therefore, the generation and identification of a nanomedicine with a highly selective TG2 inhibitor may serve as an effective approach to developing disease-modifying treatments for several neurodegenerative diseases.

## Figures and Tables

**Figure 1 ijms-25-02364-f001:**
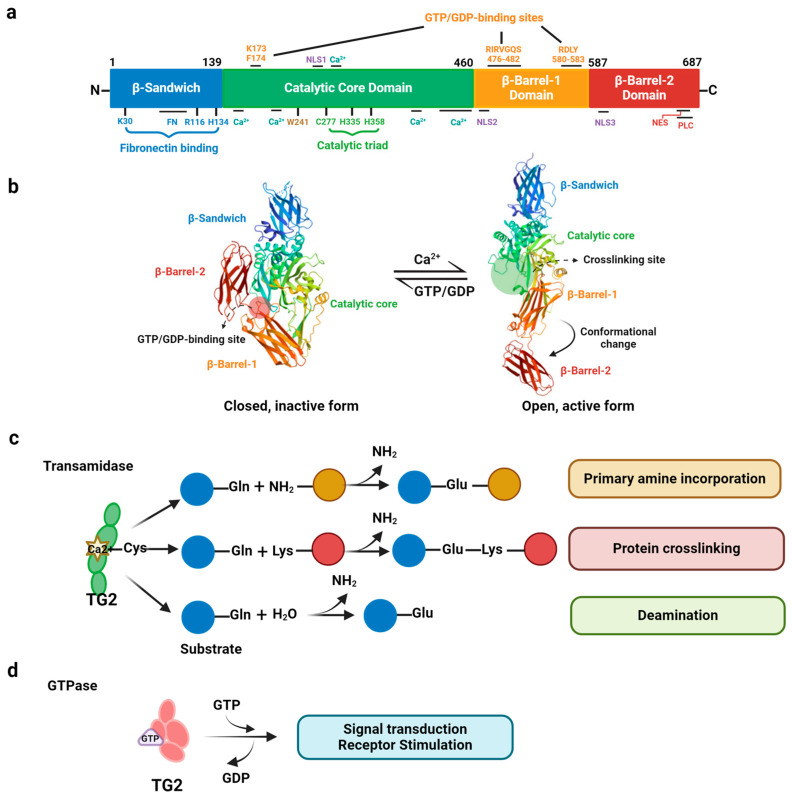
The functional domains and main biochemical activities of transglutaminase 2 (created with BioRender.com accessed on 20 December 2023). (**a**) Schematic diagram of human transglutaminase 2 (TG2) showing its four domains: β-sandwich (1–139), catalytic core (140–460), β-barrel-1 (461–586), and β-barrel-2 (587–687). The catalytic triad consisting of residues C277, H335, and D358 is responsible for the transamidation activity. Residue W241 is highly conserved among the transglutaminases and is essential to catalytic activity, likely by stabilizing the transition states. The GTP-/GDP-binding sites encompass two residues in the catalytic core domain and residues in β-barrel-1 domain. Non-canonical Ca^2+^-binding regions are indicated in the core domain, and the phospholipase C-δ1 (PLC)-binding site is shown in the β-barrel-2 domain. The nuclear localization signals, NLS1–3, and a leucine-rich nuclear export signal (NES) are indicated in the diagram. (**b**) The crystal structures of TG2 in the closed form (left, PDB ID 1KV3) and open form (right, PDB ID 2Q3Z). TG2 activity is regulated by reversible conformational changes in the catalytic core between these two forms. The closed form of TG2 is catalytically inactive as a transamidase and binds GTP/GDP (indicated as a red circle), and the C-terminal β-barrels are folded in front of the catalytic core domain. The open form of TG2 is induced by the binding of Ca^2+^ to its catalytic domain, altering the structural conformation of the enzyme by rotating the β-barrel-1 and -2 away from the catalytic domain, thus exposing the active site for crosslinking substrates (shown in the green circle). (**c**) TG2 contributes to the post translational modification of several substrate proteins via the transamidation or deamidation of specific polypeptide-bound glutamines. The transamidation reaction could mediate either the incorporation of primary amines into proteins or the crosslinking of proteins through the formation of ε-(γ-glutamyl) lysine isopeptide bonds. Blue circles indicate protein-bound Gln residues as an acyl-donor substrate of TG2, tan circles indicate primary amines, and red circles indicate Gln acceptor proteins. (**d**) TG2 exhibits GTPase activity, which is involved in regulating signal transduction.

**Figure 2 ijms-25-02364-f002:**
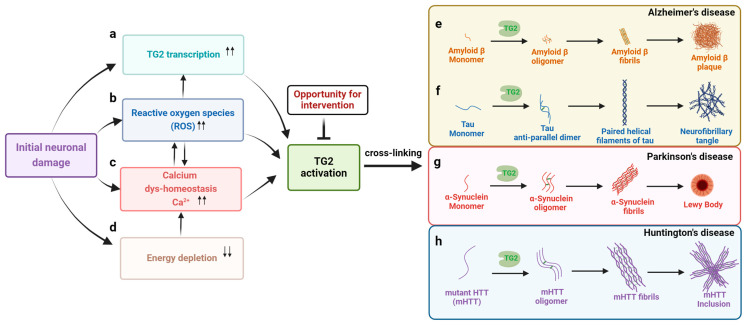
Schematic representation of transglutaminase 2 (TG2) activation and the catalyzed crosslinking of pathogenic proteins leading to the formation pathological hallmark aggregates in neurodegenerative disorders (created with BioRender.com accessed on 20 December 2023). Key factors in TG2 activation during neuronal damage include: (**a**) TG2 transcriptional upregulation leading to increased TG2 protein abundance, (**b**) oxidative stress and increased reactive oxygen species (ROS) that up-regulate TG2, especially via transcriptional control and ROS–calcium crosstalk, (**c**) the elevation of calcium levels, inducing conformational change in TG2, and (**d**) stress-induced energy depletion, which can disturb cytosolic calcium levels, triggering TG2 activation. Monomeric proteins crosslinked by activated TG2 result in the formation of toxic oligomeric species, which lead to fibrillar forms of the protein that accumulate in characteristic inclusions in the brain. (**e**) Amyloid-beta (Aβ) monomers are crosslinked together to form oligomers that subsequently form Aβ fibrils in extracellular plaques in Alzheimer’s disease. (**f**) TG2 induces the crosslinking of tau into anti-parallel dimers that proceed further into paired helical filaments and intraneuronal neurofibrillary tangles in Alzheimer’s disease and other tauopathies. (**g**) α-Synuclein oligomerization and the formation of fibrils that accumulate in intraneuronal cytoplasmic Lewy bodies and Lewy neurites in Parkinson’s disease and dementia with Lewy bodies. (**h**) TG2 catalyzes the crosslinking of mutant huntingtin (mHTT) to form insoluble fibrils in intraneuronal inclusions in Huntington’s disease.

**Figure 3 ijms-25-02364-f003:**
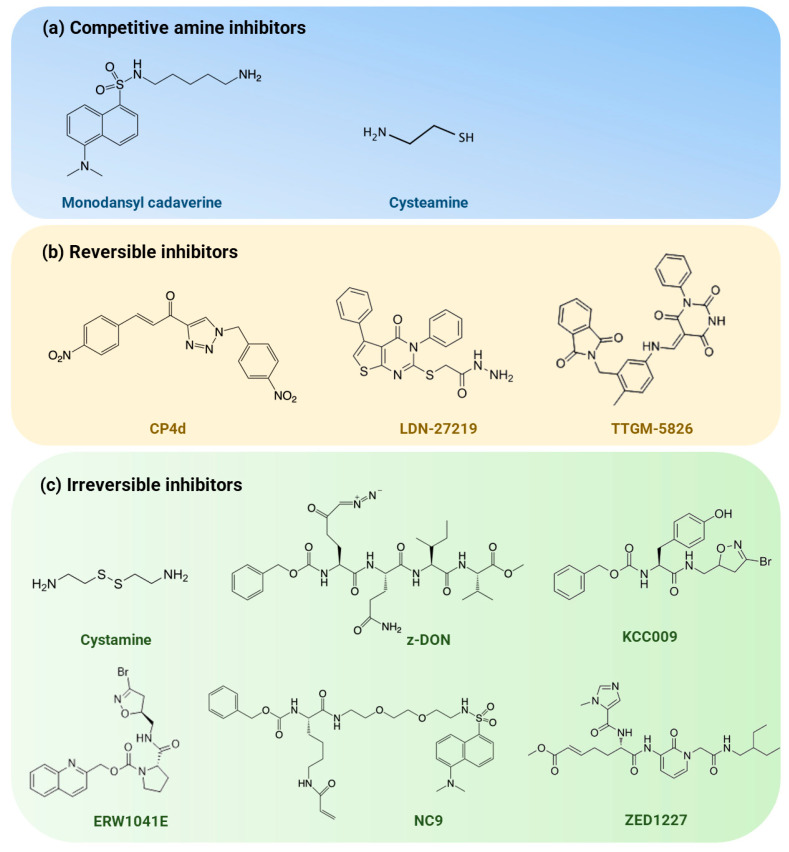
Representative inhibitors of TG2. Based on their targeting mechanism, TG2 inhibitors can be classified mainly as (**a**) competitive amine inhibitors, which compete with natural substrates for binding to the active site of TG2, preventing its crosslinking enzymatic action; (**b**) reversible inhibitors, which interact with a site on the enzyme without modifying it, inducing a conformational change that inhibits its activity; and (**c**) irreversible inhibitors, which covalently bind to the crosslinking substrate site of TG2, leading to a long-lasting inhibition. Among these inhibitors, cysteamine has been tested in phase I and II clinical trials in Huntington’s disease and ZED1227 has completed phase I and II clinical trials in celiac disease.

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
