# Peer review of "Pathogenetic Contributions and Therapeutic Implications of Transglutaminase 2 in Neurodegenerative Diseases"

_ijms, 2024, doi:10.3390/ijms25042364_

Round 1

Reviewer 1 Report

Comments and Suggestions for Authors

Reviewer comments

The present review article results show the significance of Transglutaminase (TG2) as a therapeutic target in neurological disorders. TG2 is the most ubiquitous expressed protein and shows protein crosslinking activity. It leads to aggregation of Various proteins, including Aβ, tau, α-Syn, and mHTT which is a crucial pathophysiological event in neurodegenerative disorders. This review highlighted the activity of TG2, its potent inhibitor as a therapeutic agent, and the future perspectives for the design of future highly potent and selective drugs that target TG2 in the treatment of neurodegenerative disorders (NDDs).

The paper is scientifically sound, well-planned, and written in an organized manner, and references are appropriate. However, there is a need for some improvement, as suggested below.

Scientific comments

1.      Provide a schematic representation of TG2 activation and how it leads to protein aggregation.

2.      Figure 2 is very basic and does not give information on the mechanism of TG2 working. The figure should be more informative.

3.      Has nuclear TG2 shown any epigenetic phenomenon that leads to NDDs?

4.      Discuss the mechanism of activation of TG2. Is the process/mechanism of activation of TG2 the same for all NDDs?

5.      Is prevention of TG2 activation possible? If yes, discuss briefly.

6.      Section 5, Therapeutic Implications has been discussed in very short. Discuss the few important current research data/clinical data of therapeutic inhibitors for TG2.

Minor comments

1.      Discuss this paper DOI: 10.1016/j.ab.2019.113556.

2.      Line 271, TG2 also impacts tau. Is it a four-word sentence?

3.      Check the reference style on the journal website and correct it accordingly.

Reviewer 2 Report

Comments and Suggestions for Authors

Dear Authors, 

This is a great review describing the role of TG2 in neurodegenerative diseases. My only suggestion is, in the last part, targeting TG2 for therapeutic purposes, to create a table with the potential approaches. Additionally, it would be interesting to include in this part a discussion of the benefits of this targeted therapy respect to current/trial therapies for neurodegenerative diseases. 

Thank you very much. 

Reviewer 3 Report

Comments and Suggestions for Authors

This is a review article about transglutaminase 2 (TG2) co-authored by a recognized expert in the field (M. Mouradian). The review is very thorough and well-referenced in terms of the structure of TG2, its active binding sites, and its known actions (mainly as a protein cross-linker). The article also reviews potential TG2 involvement in several neurodegenerative diseases that are all characterized by presence of cross-linked protein aggregates. This area is another area of clinical expertise of the main co-author (Mouradian), although the review is comprehensive in terms of discussing potential TG2 involvement in non-CNS conditions.

The message is indirectly conveyed that the presence of aggregated proteins is CAUSAL for neurodegenerative diseases (controversial idea which might turn out to be at least partially correct); thus, reduction of TG2-mediated protein cross linking might be therapeutic (which might also turn out to be correct). There are the intriguing few sentences about how TG2 can negatively influence the STING system (does it have any effect on cGAS activity as well?). Thus, by this scenario TG2 could reduce inflammation system activation and be therapeutic towards neurodegeneration. This possibility should  be discussed.

What is also missing is a discussion about how TG2 synthesis is regulated. There are a few mentions of scenarios where TG2 mRNA is altered, but what is known/not known about TG2 gene transcription (promoters? repressors?) and translation (any notable miRNA's involved?). This information (or lack thereof) should be part of this review article, particularly since inhibitors of TG2 activity are so comprehensively discussed.

Overall, the authors do a very admirable job of describing the structure and functions of a multi-purpose/multi-functional protein (TG2). I find no fault with what the authors have written, but feel that their review article needs expansion as discussed above.

Comments on the Quality of English Language

English language is excellent. On line 97 is one very small and easily correctible error. The word "is" is used, whereas the subject of the sentence is plural, and requires the word "are".

Round 2

Reviewer 3 Report

Comments and Suggestions for Authors

The authors have comprehensively responded to my prior criticisms.